# Fatigue Test and Unified Fatigue Life Calculation of Q460C Steel Notched Plates

**Fengjun Lv [1], Wanzhen Wang [2,\*] and Wei Zhao [1]**

1   Center of Steel Bridge, Zhejiang Institute of Communications, Hangzhou 311112, China
2   School of Civil & Environmental Engineering and Geography Science, Ningbo University,
    Ningbo 315211, China
*   Correspondence: wangwanzhen@nbu.edu.cn

**Abstract:** In the present study, a total of 20 fatigue tests on notched plates of Q460C steel were carried out, where the effects of relative stress amplitude, $\Delta\sigma/f_y$, and relative nominal maximum stress, $\sigma_{\max}/f_y$, on the fatigue life of these notched plates were carefully examined. Theoretical analyses and numerical simulations were subsequently conducted, based on an ellipsoidal fracture model originally proposed by the second author, which has been validated for use as a fracture criterion of fatigue crack, to investigate the fatigue cracking in the Q460C steel notched plates. The theoretical model was further developed to estimate the fatigue life of the Q460C steel notched plates using a unified crack growth approach originally proposed by the second author. Based on the theoretical and simulated results, the accuracy of the unified crack growth approach, and the allowable stress fatigue life formula recommended in China's code GB50017-2017, were assessed. The comparisons indicate that the unified crack growth approach is able to provide a reliable fatigue life assessment for the Q460C steel notched plates.

**Keywords:** ellipsoidal fracture model; fatigue crack; fatigue life; unified crack growth approach





## 1. Introduction

As a failure mode of metal structures, fatigue fracture occurs in many fields, such as civil engineering, mechanical engineering, railway, and aviation. According to incomplete statistics, approximately 80–90% of metal structural failures are caused by fatigue [1]. Since high-cycle fatigue does not exhibit significant macroscopic plastic deformation, sudden failures commonly result in catastrophic accidents and substantial economic losses. Thus, the fatigue performance of structural steel and its connections has become a research hotspot in building and bridge structures, machinery, and shipbuilding.

Souto et al. [2] investigated large scale Z rail section profiles, deriving S–N curves, taking into consideration the presence of residual stresses due to cold-forming. A new design S–N curve is derived from an innovative testing methodology. Božić et al. [3] presented a method for predicting fatigue crack propagation in welded stiffened panel, accounting for the effects of residual stresses, achieving good agreement with experimental results. Cheng et al. [4] carried out fatigue tests and studied the hot spot stress and fatigue behavior of SHS (square hollow section) K-joints. Fatigue cracks were observed to be initiated at the hot spots with the highest stress concentration factors, and to propagate through the plate thickness soon after initiation. Braun et al. [5] evaluated the fatigue strength of fillet-welded joints, at subzero temperatures. The measured fatigue strength was significantly higher than estimates based on international standards and data from design codes. Sedmak et al. [6] used the XFEM to analyze fatigue crack growth, fatigue life, and crack shape evolution of welded joints, taking into account different geometries and material types. Lahtinen et al. [7] studied the fatigue behavior of the MAG welds of high-strength steel. The failures in the fatigue test occurred in the recrystallized HAZ. Increasing the heat input decreased the fatigue strength of the weld. Guo et al. [8] conducted fatigue

tests on high-strength steel and its weld connections. The results corroborated that the base material of high-strength steel possesses high fatigue resistance, and the AISC360 and EC3 standard design curves are applicable for the evaluation of the butt weld fatigue performance, with adequate safety margins. Liu et al. [9] presented a model of corrosion fatigue crack propagation in S135 high-strength drill pipe steel in an H2S environment. The results revealed that the corrosion fatigue crack propagation rate curve of S135 steel is characterized by stress corrosion and corrosion fatigue. Jie et al. [10] studied the fatigue life of welded joints of steel bridges under complex stress fields and proposed a method to define the fatigue life coefficient, taking into account the stress range. Zong et al. [11] performed a study consisting of tests and numerical simulations on non-load-carrying fillets. It was found that the S–N curve design in Eurocode3 was proved to be generally suitable, but did not have sufficient safety stock for that batch of specimens. The assumption of a type I line crack, with an initial value of 0.01 mm, provided a suitable prediction, in agreement with the experimental S–N curve, with a 95% survival probability. It should be noted that the initial fatigue crack length was 0.01 mm, depending on the accuracy of the crack observation instrument. Cicero et al. [12] studied the effect of the three thermal cutting methods on the fatigue behavior of structural steel. The test results revealed that, when applied to laser boreholes, it would be a non-conservative practice to use the curve proposed by BS7608 for the borehole.

The fatigue failure of structural steel can be divided into three stages: fatigue crack initiation, stable propagation, and unstable propagation. Considering that the unstable propagation of a fatigue crack is an instantaneous fracture, the fatigue life of structural steel and its connections is generally calculated according to the sum of fatigue crack initiation life and stable propagation life. The Neuber model [13], based on the local stress–strain method, is often used for the fatigue crack initiation life. As for the stable propagation life of fatigue crack, the Paris–Erdogan model [14], based on the nominal stress method (stress amplitude), is often used. Although the fatigue failure of structural steel and its connections has accumulated rich test data, the theoretical framework to predict fatigue life has not been well solved.

This article summarizes the fatigue tests for 20 notched plates of Q460C steel, and presents a reliable fatigue life assessment method, based on a unified crack growth approach, proposed by the second author [15], for the notched plates. The effects of the relative stress amplitude, $\Delta\sigma/f_y$, and the relative nominal maximum stress, $\sigma_{max}/f_y$, on the fatigue life of the notched plates were examined. Considering an ellipsoidal fracture model proposed by the second author [16] for structural steel, as the instability propagation criterion of fatigue crack, a set of theoretical calculations and numerical simulations were established to investigate the fatigue cracking of the notched plates. Based on the theoretical and simulated results, the unified crack growth approach and the allowable stress fatigue life formula recommended in China's code GB50017-2017 [17], were then evaluated and compared.

## 2. The Problem of Fatigue Life Calculation Currently

Due to the lack of research on the definition of limit state and relevant influencing factors in the fatigue failure process, including fatigue crack initiation, stable propagation, and stable propagation, the stress amplitude criterion suggested by Equation (1) is used to calculate the fatigue life of structural steel and its connections, in China's code GB50017-2017 [17].

$$\Delta\sigma_c \leq [\Delta\sigma] = (C_z/N_f)^{1/\beta z} \tag{1}$$

where $\Delta\sigma_c$ is the converted stress amplitude, $\Delta\sigma_c = \sigma_{max} - 0.7\,\sigma_{min}$, and the dimension is MPa. $\sigma_{max}$ and $\sigma_{min}$ refer to nominal maximum and minimum stress, respectively. The parameters $C_z$ and $\beta_z$ are the coefficients related to the shape of the specimen and are dimensionless numbers. $N_f$ is a dimensionless number that denotes the fatigue life. However, there is a defect of different dimensions at both ends of the fatigue life calculation formula in this equation.

The fatigue life calculation formula suggested by Equation (1) is derived by integrating the Paris–Erdogan model [14], indicated in Equation (2).

$$\mathrm{d}a/\mathrm{d}N = B \times (\Delta K)^m \tag{2}$$

where $a$ is the length of the fatigue crack in mm. The dimensionless number $N$, is the number of load cycles. The parameter $\Delta K$, is the stress intensity factor amplitude in MPa·(mm)$^{1/2}$. $B$ and $m$ are undetermined coefficients, and $m$ is a dimensionless number.

As can be seen in Equation (2), the dimension of parameter $B$ changes with the value of dimensionless parameter $m$. One potential problem is the unit: for example, when $m = 2$, the unit of $B$ is MPa$^{-2}$, while it is MPa$^{-4}$·mm$^{-1}$ when $m = 4$.

## 3. Ellipsoidal Fracture Model and Unified Crack Growth Approach

### 3.1. Ellipsoidal Fracture Model of Structural Steel

Base on the assumption that the fracture strain reaches a minimum value for metals under equal triaxial tensions with the stress triaxiality, $\sigma_\mathrm{m}/\sigma_\mathrm{seq} \approx \infty$, Wang [16] proposed an ellipsoidal fracture model and coupled yield model for structural steel:

$$(\sigma_\mathrm{seq}/r)^2 + (\sigma_\mathrm{m}/q)^2 = 3\tau_\mathrm{y}{}^2 \tag{3}$$

$$\sigma_\mathrm{seq}{}^2 + (\sigma_\mathrm{m}/q)^2 = 3\tau_\mathrm{y}{}^2 \tag{4}$$

where $q = \frac{\sqrt{2}(1+\mu)}{3(1-2\mu)}$, $\sqrt{3}\tau_\mathrm{y} = \frac{\sqrt{1+9q^2}}{3q} f_\mathrm{y}$, and $r = \tau_\mathrm{f}/\tau_\mathrm{y}$. $\sigma_\mathrm{seq}$ and $\sigma_\mathrm{m}$ are the von Mises equivalent stress and mean stress, respectively. The parameters $\tau_\mathrm{f}$, $\tau_\mathrm{y}$, $f_\mathrm{y}$, and $\mu$ denote the shear fracture strength, shear yield strength, tensile yield strength, and the Poisson's ratio. The parameter $r$ can be derived from $\frac{f_\mathrm{u}}{f_\mathrm{y}} \approx \frac{f_\mathrm{f}}{f_\mathrm{y}} = \frac{r\sqrt{1+9q^2}}{\sqrt{r^2+9q^2}}$. The parameters $f_\mathrm{u}$ and $f_\mathrm{f}$ represent the material's uniaxial ultimate strength and uniaxial fracture strength, respectively.

### 3.2. Unified Crack Growth Approach of Structural Steel

Based on the fact that the growth of fatigue cracks accelerates with the number of load cycles, $N$, Wang [15] assumed that the growth rate of fatigue crack initiation and stable propagation, $v_\mathrm{c}$, increases monotonically as a power function:

$$v_\mathrm{c} = b \times N^{\,c} + d \tag{5}$$

where $b$, $c$, and $d$ are undetermined coefficients.

From the initial condition: $v_{\mathrm{c}\,|\,N=0} = 0$, we get: $d = 0$. Then Equation (5) is simplified as follows,

$$v_\mathrm{c} = b \times N^{\,c} \tag{6}$$

Integrate Equation (6) to obtain Equation (7).

$$a_\mathrm{f} = N_\mathrm{f}{}^{c+1} \times b/(c + 1) \tag{7}$$

Let $\xi = b/(c + 1)$ and $\eta = c + 1$, and Equation (7) is simplified as:

$$a_\mathrm{f} = \xi \times N_\mathrm{f}{}^{\eta} \tag{8}$$

To calculate the fatigue crack initiation and stable propagation life, a theoretical model based on Equation (8) was originally proposed by the second author [15] as follows,

$$N_\mathrm{f} = (a_\mathrm{f}/\xi)^{1/\eta} \tag{9}$$

where $\eta$ is the reciprocal of the exponent (unitless) in the power function of the fatigue life $N_\mathrm{f}$, and the fatigue crack initiation and stable propagation length $a_\mathrm{f}$. The physical meaning

of parameter $\xi$ is a measurement related to the fatigue crack growth mode. When $N_f$ and $a_f$ follow the power function evolution with index $1/\eta$, the dimension of parameter $\xi$ is mm/times$^\eta$. In particular, when $\eta = 1$, the dimension of parameter $\xi$ is mm/times, and the physical meaning is fatigue crack initiation and stable growth rate. When $\eta = 2$, the dimension of parameter $\xi$ is mm/times$^2$, and its physical meaning is fatigue initiation and stable crack growth acceleration. In this way, the dimension of $(a_f/\xi)^{1/\eta}$ is the time, following the same dimension as another side in this equation. Therefore, the problem of the dimension issue in Equations (1) and (2) is resolved in Equation (9).

The framework for the unified calculation of fatigue crack initiation life and stable propagation life proposed by the second author [15] is addressed below:

(1) The fatigue fracture area, $A$, of structural steel and its connections is the sum of fatigue crack initiation and stable propagation area, $A_f$ (including initial defect effect), and fatigue crack instability propagation area, $A_n$ (tensile fracture area under the nominal maximum stress), i.e., $A = A_f + A_n$.

(2) The ellipsoidal fracture model of structural steel, originally proposed by the second author [16], is used as the cracking criterion of the crack tip during fatigue crack instability propagation, and can be expressed as Equation (3).

(3) Assuming that the fatigue crack starts at the initial crack or defect and propagates radially to the periphery, the functions between fatigue crack initiation and stable propagation length, $a_f$, and fatigue crack initiation and stable propagation area, $A_f$, are obtained. See Section 5.1 for the calculation process.

(4) Since the fatigue crack growth accelerates with the number of load cycles, $N$, it is assumed that the fatigue crack initiation and stable growth rate, $v_c$, is a monotonically increasing power function of the number of load cycles, $N$, as shown in Equation (5).

(5) As Equation (7) suggests, a fatigue life calculation model that uniformly calculates the fatigue crack initiation and stable propagation life is obtained by integrating the fatigue crack growth rate equation.

It should be noted that the fatigue life prediction model suggested by Equation (9) is dependent on the ellipsoidal fracture model suggested by Equation (3), as the cracking criterion of the crack tip. The application premise of the ellipsoidal fracture model suggested by Equation (3) is the average stress $\sigma_m = (\sigma_1 + \sigma_2 + \sigma_3)/3 \geq 0$. Therefore, the fatigue life calculation model suggested by Equation (7) is only applicable to the cyclic loading condition where the nominal maximum stress $\sigma_{max} \geq 0$. This explains that fatigue failure will not occur when structural steel and its connections bear no tensile stress cycle.

The fatigue life calculation model proposed in Equation (9) indicates that, when the stress field at the initial defect or the crack tip of the initial crack, i.e., does not reach the ellipsoidal fracture model proposed in Equation (3), the initial crack cannot be formed, resulting in an infinite fatigue life scenario. This explains the fatigue limit phenomenon of structural steel and its connections.

## 4. Fatigue Tests on Q460C Steel Notched Plates

*Specimen Design*

Figure 1 shows the geometry of the specimens, consisting of 20 notched plates of Q460C steel, specified in China's code GB/T 3075-2008 [18]. The thickness of the notched plates, $t$, remains fixed at 4 mm for all specimens. The notched radius, $r_2$, equals 2 mm for all specimens, to fix fatigue crack initiation at the root of the notch and to exclude the influence of different crack initiation positions on fatigue life of the Q460C steel plate. $w_0$ is the width of the notched section. Figure 2 shows pictures of the Q460C steel notched plates. Table 1 lists the dimensions and loading parameters of the specimens.

Specimens A1–A4 were designed to study the effects of the stress amplitude, $\Delta\sigma = \sigma_{max} - 0.7\,\sigma_{min}$, on the fatigue life of the Q460C steel notched plates. Specimens B1–B16 were designed to investigate the effects of the nominal maximum stress, $\sigma_{max}$, and the stress amplitude, $\Delta\sigma$, on the fatigue life of the Q460C steel notched plates.

Material parameters of the Q460C steel were tested according to China's code GB/T 228.1-2010 [19]. Table 2 lists the corresponding mechanical properties of the Q460C steel.

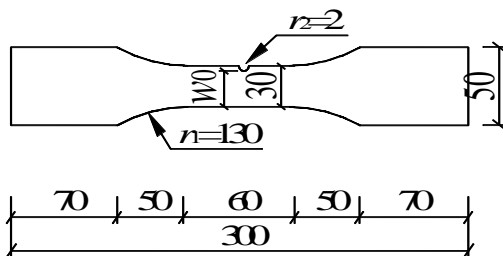

**Figure 1.** Geometry of the Q460C steel notched plates.

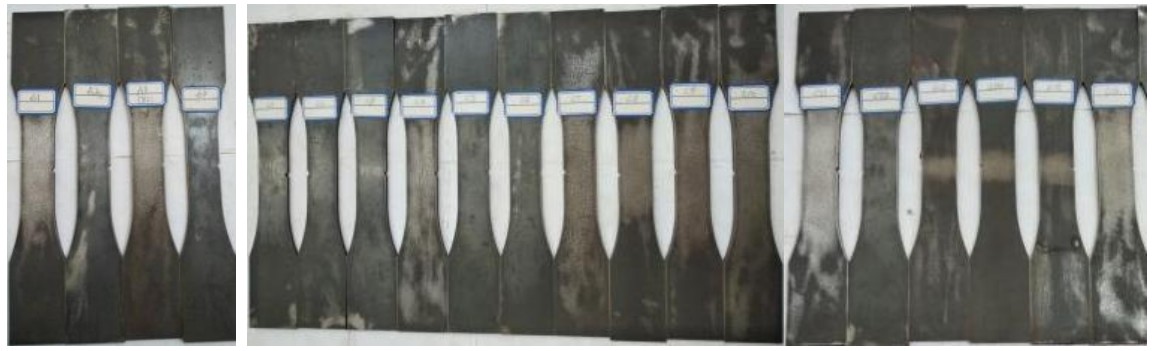

**Figure 2.** Pictures of the Q460C steel notched plates.

**Table 1.** Geometry and loading parameters of the Q460C steel notched plates.

| Specimen No. | $w_0$ (mm) | $t$ (mm) | $A$ (mm$^2$) | $\sigma_{max}/f_y$ | $\sigma_{min}/f_y$ | $\Delta\sigma_c/f_y$ | $\Delta\sigma/f_y$ |
|---|---|---|---|---|---|---|---|
| A1 | 28.1 | 4.1 | 115.21 | 0.70 | 0.29 | 0.50 | 0.41 |
| A2 | 28.3 | 4.1 | 116.03 | 0.60 | 0.14 | 0.50 | 0.46 |
| A3 | 28.2 | 4.1 | 115.62 | 0.50 | 0.00 | 0.50 | 0.50 |
| A4 | 28.1 | 4.1 | 115.21 | 0.40 | −0.14 | 0.50 | 0.54 |
| **Specimen No.** | **$w_0$ (mm)** | **$t$ (mm)** | **$A$ (mm$^2$)** | **$\sigma_{max}/f_y$** | **$\sigma_{min}/f_y$** | **$\Delta\sigma/f_y$** | |
| B1 | 28.1 | 4.1 | 115.21 | 0.80 | 0.10 | 0.70 | |
| B2 | 28.2 | 4.0 | 112.80 | 0.70 | 0.00 | 0.70 | |
| B3 | 28.3 | 4.1 | 116.03 | 0.60 | −0.10 | 0.70 | |
| B4 | 28.1 | 4.1 | 115.21 | 0.50 | −0.20 | 0.70 | |
| B5 | 28.2 | 4.0 | 112.80 | 0.70 | 0.10 | 0.60 | |
| B6 | 28.1 | 4.1 | 115.21 | 0.60 | 0.00 | 0.60 | |
| B7 | 28.2 | 4.0 | 112.80 | 0.50 | −0.10 | 0.60 | |
| B8 | 28.1 | 4.0 | 112.40 | 0.40 | −0.20 | 0.60 | |
| B9 | 28.2 | 4.0 | 112.80 | 0.70 | 0.20 | 0.50 | |
| B10 | 28.1 | 4.1 | 115.21 | 0.60 | 0.10 | 0.50 | |
| B11 | 28.1 | 4.0 | 112.40 | 0.50 | 0.00 | 0.50 | |
| B12 | 28.2 | 4.1 | 115.62 | 0.40 | −0.10 | 0.50 | |
| B13 | 28.2 | 4.1 | 115.62 | 0.70 | 0.30 | 0.40 | |
| B14 | 28.1 | 4.1 | 115.21 | 0.60 | 0.20 | 0.40 | |
| B15 | 28.1 | 4.1 | 115.21 | 0.50 | 0.10 | 0.40 | |
| B16 | 28.1 | 4.0 | 112.40 | 0.40 | 0.00 | 0.40 | |

**Table 2.** Tested material properties of the Q460C steel.

| Yield Strength, $f_y$ | Ultimate Strength, $f_u$ | Yield Strain, $\varepsilon_y$ | Ultimate Strain, $\varepsilon_u$ | $E$ | $\mu$ |
|---|---|---|---|---|---|
| (MPa) | (MPa) | (%) | (%) | (GPa) | |
| 540.8 | 629.0 | 0.032 | 14.0 | 202.6 | 0.28 |

Figure 3 shows the test setup and the fractured specimens after the fatigue test. The specimen was mounted between two end fixtures attached to a 500 kN electro-hydraulic servo fatigue tester. One end of the notched plate was fixed, while the other end was loaded by a force controlled loading. The loading waveform was a constant amplitude sinusoid, and the loading frequency fluctuated at a range of 70−80 Hz.

The test results show that the fatigue crack initiated at the edge of the notch and propagated along the width of plate at the notched section, to the outer edge of the plate, and penetrated the thickness of plate. A fatigue fracture occurred at the notched section when the fatigue crack penetrated the width of the notched section.

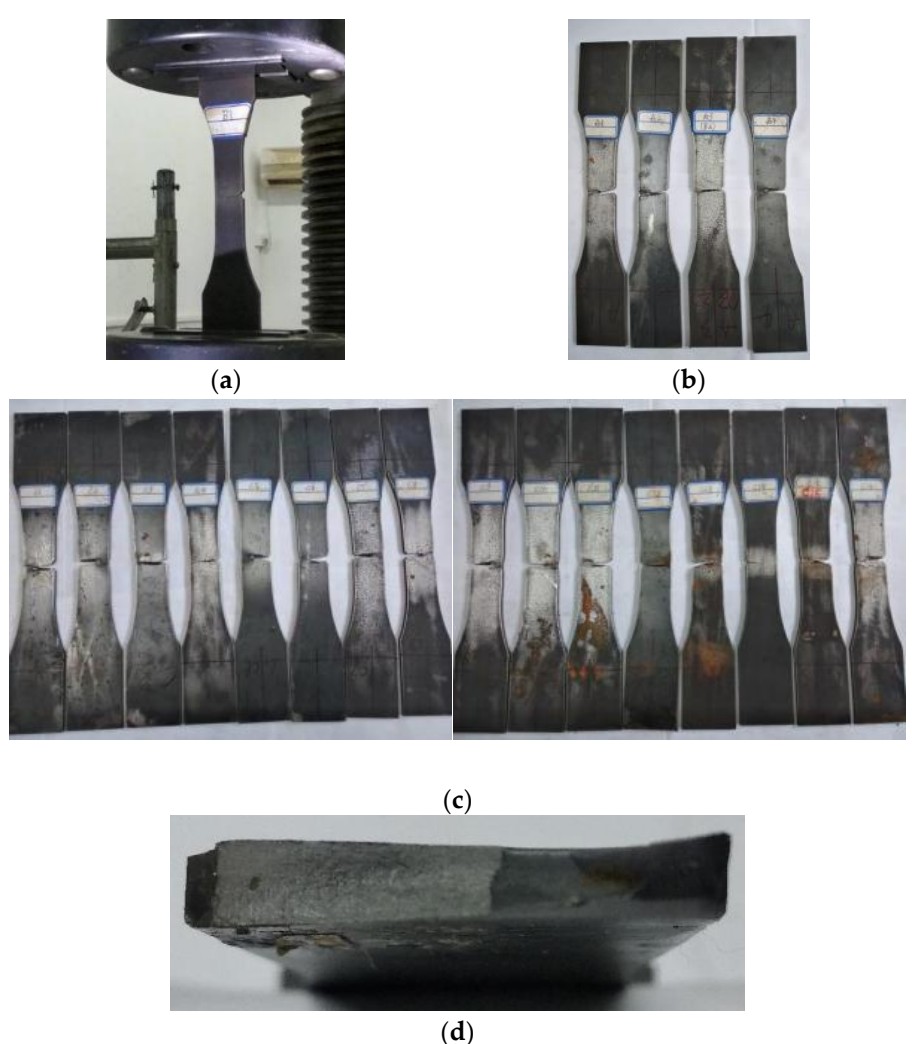

(a)

(b)

(c)

(d)

**Figure 3.** Test setup and fatigue fracture of the Q460C steel notched plates. (**a**) Test setup, (**b**) specimens A1–A4, (**c**) specimens B1–B16, (**d**) fatigue fracture.

The tested fatigue lives of the Q460C steel notched plates are listed in Table 3, in which $N_{f,s}$ is the fatigue life calculated by Equation (1) in China's code GB50017-2017 [17] (specimen type Z4: $C_Z = 2.81 \times 10^{12}$, $\beta_Z = 3$).

The test results of specimens A1–A4 reveal that the relative stress amplitude, $\Delta\sigma/f_y$, rather than the relative converted stress amplitude, $\Delta\sigma_c/f_y$, is the stress parameter that affects the fatigue life of the notched plates. The fatigue life of the notched plates increases when the relative stress amplitude, $\Delta\sigma/f_y$, and the relative nominal maximum stress, $\sigma_{max}/f_y$, decrease. The effect of the relative stress amplitude, $\Delta\sigma/f_y$, on the fatigue life of the notched plate is larger than that of the relative nominal maximum stress, $\sigma_{max}/f_y$.

The fatigue life predictions determined using the fatigue life formula specified in China's current code GB50017-2017 [17], are conservative for the specimens with high stress amplitude (i.e., $\Delta\sigma \geq 0.5f_y$), while being unsafe for those with low stress amplitude (i.e., $\Delta\sigma = 0.4f_y$), with the calculation errors, $e_{s\text{-}t} = (N_{f,s} - N_{f,t})/N_{f,t} = -17.1\%$ to $+84.9\%$. The calculated fatigue life of specimen B1, with the most significant calculation error, is 1.85 times the test value.

The effect of the mean stress of the loading cycle is not taken into account in Equation (1), resulting in the high errors registered in Table 3, especially those with the "+" sign, which are on the unsafe side. Equation (1) is valid for a symmetrical (fully reversed) cycle $R = \sigma_{min}/\sigma_{max} = -1$.

**Table 3.** Fatigue test results of the Q460C steel notched plates.

| Specimen No. | $\Delta\sigma_c/f_y$ | $\Delta\sigma/f_y$ | $N_{f,t}$ (cycles) | $N_{f,s}$ (cycles) | $e_{s\text{-}t}$ (%) | $\Delta\sigma_{G,SC}/f_y$ | $N_{f,G}$ (cycles) | $e_{G\text{-}t}$ (%) |
|---|---|---|---|---|---|---|---|---|
| A1 | 0.50 | 0.41 | 171,400 | 142,130 | −17.1 | 0.26 | 260,794 | +52.2 |
| A2 | 0.50 | 0.46 | 135,700 | 142,130 | +4.7 | 0.34 | 121,193 | −10.7 |
| A3 | 0.50 | 0.50 | 129,700 | 142,130 | +9.6 | 0.41 | 68,850 | −43.7 |
| A4 | 0.50 | 0.54 | 113,600 | 142,130 | +25.1 | 0.48 | 43,192 | −62.0 |
| **Specimen No.** | $\Delta\sigma/f_y$ | $\sigma_{max}/f_y$ | $N_{f,t}$ (cycles) | $N_{f,s}$ (cycles) | $e_{s\text{-}t}$ (%) | $\Delta\sigma_{G,SC}/f_y$ | $N_{f,G}$ (cycles) | $e_{G\text{-}t}$ (%) |
| B1 | 0.70 | 0.80 | 24,700 | 45,670 | +84.9 | 0.37 | 92,952 | +276.3 |
| B2 | 0.70 | 0.70 | 31,600 | 51,798 | +63.9 | 0.45 | 52,403 | +65.8 |
| B3 | 0.70 | 0.60 | 36,000 | 59,071 | +64.1 | 0.51 | 34,392 | −4.5 |
| B4 | 0.70 | 0.50 | 44,900 | 67,773 | +50.9 | 0.57 | 25,091 | −44.1 |
| B5 | 0.60 | 0.70 | 49,100 | 71,052 | +44.7 | 0.38 | 83,214 | +69.5 |
| B6 | 0.60 | 0.60 | 57,200 | 82,251 | +43.8 | 0.44 | 54,613 | −4.5 |
| B7 | 0.60 | 0.50 | 67,700 | 95,934 | +41.7 | 0.49 | 39,844 | −41.1 |
| B8 | 0.60 | 0.40 | 87,800 | 112,828 | +28.5 | 0.53 | 31,487 | −64.1 |
| B9 | 0.50 | 0.70 | 90,700 | 101,165 | +11.5 | 0.32 | 143,794 | +58.5 |
| B10 | 0.50 | 0.60 | 103,300 | 119,335 | +15.5 | 0.37 | 94,371 | −8.6 |
| B11 | 0.50 | 0.50 | 129,700 | 142,130 | +9.6 | 0.41 | 68,850 | −46.9 |
| B12 | 0.50 | 0.40 | 157,400 | 171,121 | +8.7 | 0.44 | 544,09 | −65.4 |
| B13 | 0.40 | 0.70 | 178,300 | 151,011 | −15.3 | 0.26 | 280,847 | +57.5 |
| B14 | 0.40 | 0.60 | 219,900 | 182,525 | −17.0 | 0.29 | 184,319 | −16.2 |
| B15 | 0.40 | 0.50 | 253,000 | 223,455 | −11.7 | 0.33 | 134,472 | −46.8 |
| B16 | 0.40 | 0.40 | 319,700 | 277,598 | −13.1 | 0.35 | 106,268 | −66.8 |

The Gerber ruler, for taking into account the mean stress, can be used for an unsymmetrical cycle.

$$\Delta\sigma_G = \Delta\sigma \times (1 - (\sigma_{max}/f_u)^2) \tag{10}$$

The stress concentration in the Q460C steel notched plates is not considered in Equation (10). Figure 4 shows the first principle stress calculated numerically, $\sigma_{1,ns}$, at the notched section under the maximum fatigue load, $P_{max}$, for specimen B1, in which $\sigma_{1,nom} = P_{max}/(w_0 \times t)$, and the abscissa 0 and 1 represent the crack tip of the fatigue crack and the outer edge of the notched plate, respectively. See Section 5.2 for the finite element model, boundary conditions, and fatigue load for specimen B1. It can be seen in Figure 4

that the fatigue crack tip forms a high stress concentration. The stress concentration factor is calculated as: $F_{sc} = \sigma_{1,max}/\sigma_{1,av} = 1.562$, where $\sigma_{1,av}$ is the average first principle stress calculated numerically at the notched section.

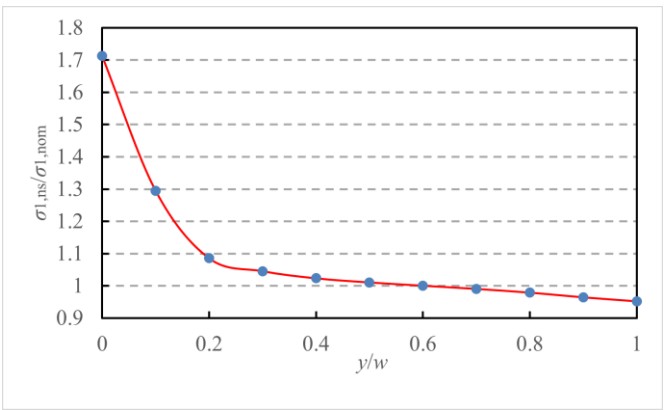

**Figure 4.** The distribution of the first principle stress calculated numerically at the notched section, under the maximum fatigue load, $P_{max}$, for specimen B1.

The modified Gerber ruler, for taking into account the stress concentration, is as follows,

$$\Delta\sigma_{G,SC} = F_{sc} \times \Delta\sigma_G = 1.562 \times \Delta\sigma \times (1 - (\sigma_{max}/f_u)^2) \tag{11}$$

Substituting Equation (11) into Equation (1), the fatigue lives calculated by the modified Gerber ruler, $N_{f,G}$, and the relative errors with the test fatigue lives, $e_{G-t} = (N_{f,G} - N_{f,t})/N_{f,t}$, are listed in Table 3. The fatigue life predictions determined using the modified Gerber ruler are unsafe for the specimens with high nominal maximum stress (i.e., $\sigma_{max} \geq 0.7f_y$), while being conservative for those with low maximum stress (i.e., $\sigma_{max} < 0.7f_y$), with the calculation errors being from $-66.8\%$ to $+276.3\%$. For specimens with large nominal maximum stresses, the stress amplitudes calculated from the Gerber ruler are reduced too much, resulting in large fatigue lives.

## 5. Calculation of Fatigue Crack Initiation and Stable Propagation Length

The parameters $\xi$ and $\eta$ of the fatigue life calculation model suggested by Equation (9), are calibrated by fatigue life, $N_f$, and fatigue crack initiation and stable propagation length, $a_f$. The fatigue life, $N_f$, is obtained from the fatigue test, and the fatigue crack initiation and stable propagation length, $a_f$, is calculated according to the following method.

### 5.1. Theoretical Estimation of Fatigue Crack Initiation and Stable Propagation Length

According to the fatigue failure mode of the notched plates shown in Figure 3, the fatigue crack initiates at the edge of the notch, and the fatigue fracture occurs at the notched section.

The first principle stress, $\sigma_1$, along the length of plate, at the unstable propagation section of the fatigue crack in the notched plate is,

$$\sigma_1 = P_{max}/A_{n,t} = \sigma_{max} \times A/A_{n,t} \tag{12}$$

where $P_{max}$ and $\sigma_{max}$ are the maximum fatigue load and the nominal maximum stress, respectively. $A$ is the fatigue fracture area of the notched plate, $A = w_0 \times t$. $w_0$ and $t$ are the width of the notched section and the thickness of the notched plate, respectively. $A_{n,t}$ is the unstable propagation area of the fatigue crack, calculated theoretically.

It is conservatively assumed that the direction of the width of the plate is fixed at the unstable propagation section of the fatigue crack in the notched plate. The second principle

stress, $\sigma_2$, along the width of the plate, at the unstable propagation section of the fatigue crack in the notched plate is,

$$\sigma_2 \approx \mu \times \sigma_1 \tag{13}$$

where $\mu$ is the Poisson's ratio of the Q460C steel.

The fatigue crack penetrates the thickness of the notched plate, shown in Figure 3, the third principle stress, $\sigma_3$, along the thickness of the plate at the unstable propagation section of the fatigue crack in the notched plate, is expressed as: $\sigma_3 = 0$.

The von Mises equivalent stress, $\sigma_{\mathrm{seq}}$, and mean stress, $\sigma_{\mathrm{m}}$, at the unstable propagation section of the fatigue crack are as follows:

$$\sigma_{\mathrm{seq}} = \sqrt{1 - \mu + \mu^2}\sigma_{\max} \cdot A / A_{\mathrm{n,t}} \tag{14}$$

$$\sigma_{\mathrm{m}} = (\sigma_1 + \sigma_2 + \sigma_3)/3 = (1 + \mu)\sigma_{\max} \times A/3A_{\mathrm{n,t}} \tag{15}$$

Substituting Equations (14) and (15) into Equation (3), the unstable propagation area of the fatigue crack in the notched plate is obtained,

$$A_{\mathrm{n,t}} = \frac{\sqrt{9q^2(1 - \mu + \mu^2) + (1 + \mu)^2 r^2} \cdot \sigma_{\max} A}{3\sqrt{3}qr \cdot \tau_{\mathrm{y}}} \tag{16}$$

The initiation and stable propagation area, $A_{\mathrm{f,t}}$, and the initiation and stable propagation length, $a_{\mathrm{f,t}}$, of the fatigue crack, calculated theoretically, are expressed as:

$$A_{\mathrm{f,t}} = A - A_{\mathrm{n,t}} \tag{17}$$

$$a_{\mathrm{f,t}} = A_{\mathrm{f,t}}/t - a_0 \tag{18}$$

where $a_0$ is the size of the initial defect, $a_0 = 0$.

### 5.2. Numerical Calculation of Fatigue Crack Initiation and Stable Propagation Length

5.2.1. Implementation of the Ellipsoidal Yield Model in ANSYS

To incorporate the effect of the mean stress on the yielding and fracture failure for the Q460C steel, this section implements the ellipsoidal yield model, suggested by Equation (4), in the ANSYS Parametric Design Language (APDL) and User Programmable Features (UPFs). The numerical procedure then computes the stress fields at the fatigue fracture section of the Q460C steel notched plate under the maximum fatigue load, $P_{\max}$, using this ellipsoidal yield model. It should be noted that, implementing the ellipsoidal yield model in the numerical procedure depends on the sign of the mean stress of the material. For materials or elements with zero or positive mean stress, $\sigma_{\mathrm{m}} \geq 0$, the numerical procedure mobilizes the ellipsoidal yield model. In contrast, for materials or components with negative mean stress, $\sigma_{\mathrm{m}} < 0$, the finite element solver uses the von Mises yield model available in ANSYS. The current implementation of the ellipsoidal yield model assumes isotropic material properties. The procedure to program the ellipsoidal yield model using the User Programmable Features (UPFs), through the source code of ANSYS (version 8.1), originally proposed by the second author [20] is detailed as follows:

(1)  Use FORTRAN language to create or modify the programming code for the ellipsoidal yield model and the flow rule subroutines;
(2)  Run "anscust.bat" to compile the new code and generate the new "ANSYS.EXE";
(3)  Use ANSYS Parametric Design Language to plant the new code into the ANSYS program.

5.2.2. Flow Rule and Hardening Law

The incremental plastic strain tensor, $d\varepsilon_{\mathrm{ij}}{}^{\mathrm{p}}$, describes the plastic deformation for an elastic-plastic metallic material. A complete description of the incremental plastic strain

tensor requires the determination of: (1) the ratio among different components of the plastic strain increment; and (2) their magnitudes concerning the stress increment, $d\sigma_{ij}$. The flow rule defines the relationship between the next increment of the plastic strain increment, $d\varepsilon_{ij}^{P}$, and the present state of stress, $d\sigma_{ij}$, for a material point subjected to further loading. Based on the classical plasticity theory, the plastic potential function, g, for metallic materials assumes the same shape as the yield function $f$, i.e., $g = f$. This type of flow rule is namely the associated flow rule, and is given by:

$$d\varepsilon_{ij}^{P} = d\lambda\, \partial g / \partial\sigma_{ij} = d\lambda\, \partial f / \partial\sigma_{ij} \tag{19}$$

where $d\lambda$ is a positive scalar of proportionality dependent on the current state of stress and loading history. The generalized yield function $f$, is defined by Equation (9).

The rule governing the evolution of a loading surface is called the hardening law. The isotropic hardening rule inherent in ANSYS is adopted. The isotropic hardening rule assumes that the yield surface expands uniformly, without distortion and translation, during a loading process. The yield surface for anisotropic hardening material is generally expressed by:

$$f(\sigma_{ij}, \kappa) = f_0(\sigma_{ij}) - k(\kappa) = 0 \tag{20}$$

where $k(\kappa)$, a hardening or growth function, defines the size of the surface, and $\kappa$ is a hardening parameter whose value represents the plastic loading history of the high-strength steel.

The accuracy of the ellipsoid yield model, and its flow rule and hardening law, was verified by the numerical simulation of the tensile fracture test of a notched bar, carried out by the second author [20].

5.2.3. Material Parameters

The material properties of Q460C steel are obtained from the material property test data listed in Table 2. The numerical procedure adopts the following material properties of Q460C steel: $f_y$ = 540.8 MPa, $f_u$ = 629.0 MPa, $E$ = 202.6 GPa, and $\mu$ = 0.28.

The ellipsoidal fracture model of Q460C steel is quantified as follows: $q = \frac{\sqrt{2}(1+\mu)}{3(1-2\mu)} \approx$ 1.37, $\sqrt{3}\tau_y = \frac{\sqrt{1+9q^2}}{3q} f_y \approx$ 556.6 MPa, and $r \approx$ 1.18, which was derived from $\frac{f_u}{f_y} \approx \frac{r\sqrt{1+9q^2}}{\sqrt{r^2+9q^2}}$.

5.2.4. Geometric Model, Boundary Conditions, and Fatigue Load

As previously mentioned, the fatigue crack initiates at the edge of the notch and propagates along the width of the plate at the notched section, to the outer edge of the plate. The fatigue fracture occurs at the notched section when the fatigue crack penetrates the width of the notched section, shown in Figure 3. Figure 5 shows the typical finite element meshes built from 20-node 3D prismatic elements, or Solid95, in the ANSYS element library, for notched plate specimen B1. The feasibility of the Solid95 element for the stress singularity at the crack tip has been verified in the literature [20]. The finite element model adopts finer meshes near the crack tip and relatively coarse meshes away from the crack tip. The model shown in Figure 5 contains 16584 elements and 17632 nodes, with the smallest element size being 0.2–0.5 mm. The geometric dimensions of the finite element model of specimen B1 are the same as that of the fatigue test.

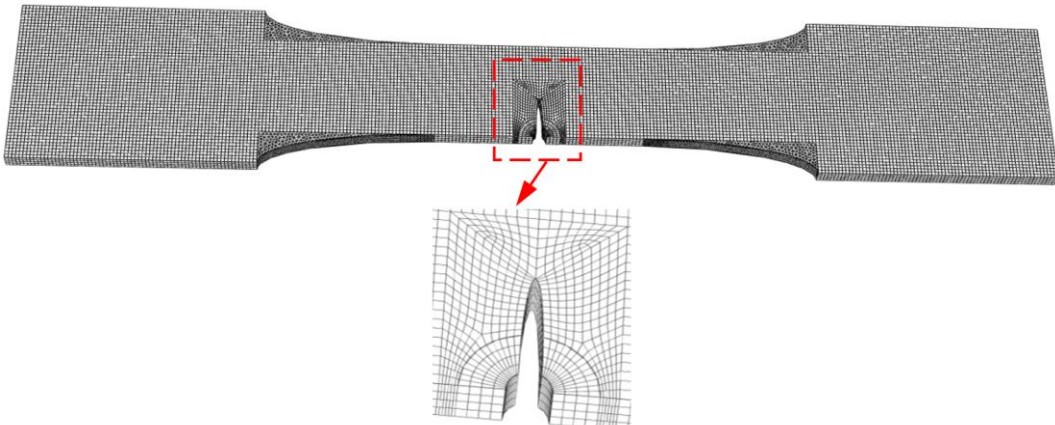

**Figure 5.** Finite element meshes of specimen B1.

In order to easily simulate cracks penetrating the thickness of the notched plate in the ANSYS software (version 9.0), a cylindrical crack with a semi-elliptic cross section, rather than a simpler flat crack shape, was introduced at the edge of the notch. The height of the cylinder is parallel to the thickness of the notched plate. The cylinder passes through the thickness of the notched plate to simulate fatigue crack penetration through the thickness of the notched plate. The intersection area of the semi-elliptical cylinder and the notched plate is $A_{f,t}$. The major axis of the ellipse is parallel to the width of the notched plate, and the minor axis is parallel to the length of the notched plate. The lengths of the semimajor axis and semiminor axis of the ellipse are $a_c = a_{f,t} + a_0$ and $b_c = 0.1a_c$, respectively, that is, $b_c/a_c \ll 1$. This is used to simulate the fatigue crack with propagation length, $a_{f,t}$, and propagation area, $A_{f,t}$, calculated theoretically.

The loading and boundary of the notched plate in the finite element simulation resemble the conditions in the fatigue test, where one end of the notched plate is fixed and the maximum fatigue load, $P_{max}$, is applied at the other end. The ellipsoidal fracture model suggested by Equation (3) is used as the crack criterion of the fatigue crack tip and unstable propagation (fracture) of a fatigue crack. The "kill" element function in the ANSYS software, is used to simulate fatigue crack cracking and stress release.

Considering the stress release effect at the crack tip of the fatigue crack, it is conservatively agreed in this paper that when the average stress field on the residual effective area of the notched plate reaches the ellipsoidal fracture model suggested by Equation (3), the residual effective area of the fillet weld at this time is defined as the unstable propagation area of a fatigue crack. Hence, the numerically calculated average equivalent stress and mean stress at the unstable propagation area, $A_{n,t}$ (theoretical calculation), of the fatigue crack are substituted into the ellipsoidal fracture model suggested by Equation (3).

If $(\overline{\sigma}_{seq,2}/r)^2 + (\overline{\sigma}_{m,2}/q)^2 < 3\tau_y^2$, taking the crack length at the crack tip as the step, the semimajor axis of the elliptical crack is gradually increased from $a_{f,t} + a_0$ to $a_{f,i} + a_0$. The crack length is the length of the region where the stress field at the crack tip reaches the ellipsoidal fracture model suggested by Equation (3). The fatigue crack initiation and stable propagation area increase from $A_{f,t}$ to $A_{f,i}$, and the fatigue crack unstable propagation area decreases from $A_{n,t}$ to $A_{n,i}$. The finite element model of the fatigue test specimen is re-established until the average equivalent stress $\overline{\sigma}_{seq,i}$ and mean stress $\overline{\sigma}_{m,i}$ at the unstable propagation area of the fatigue crack, calculated numerically, meet: $(\overline{\sigma}_{seq,i}/r)^2 + (\overline{\sigma}_{m,i}/q)^2 \approx 3\tau_y^2$.

If $(\overline{\sigma}_{seq,2}/r)^2 + (\overline{\sigma}_{m,2}/q)^2 > 3\tau_y^2$, taking the crack length at the crack tip as the step, the semimajor axis of the elliptical crack is gradually decreased from $a_{f,t} + a_0$ to $a_{f,i} + a_0$. The fatigue crack initiation and stable propagation area decrease from $A_{f,t}$ to $A_{f,i}$, and the fatigue crack unstable propagation area increases from $A_{n,t}$ to $A_{n,i}$. The finite element model of the fatigue test specimen is re-established until the average equivalent stress $\overline{\sigma}_{seq,i}$ and mean

stress $\overline{\sigma}_{m,i}$ at the unstable propagation area of the fatigue crack, calculated numerically, meet: $(\overline{\sigma}_{seq,i}/r)^2 + (\overline{\sigma}_{m,i}/q)^2 \approx 3\tau_y^2$.

Figure 6 shows the numerically calculated stress field during unstable propagation of the fatigue crack in specimen B1, in which the abscissa 0 and 1 represent the crack tip of the fatigue crack and the outer edge of the notched plate, respectively. Additionally, the cracking index is calculated by the ellipsoidal fracture model suggested by Equation (3) as:

$$I_C = \sqrt{(\sigma_{seq}/r)^2 + (\sigma_m/q)^2}/\sqrt{3}\tau_y.$$

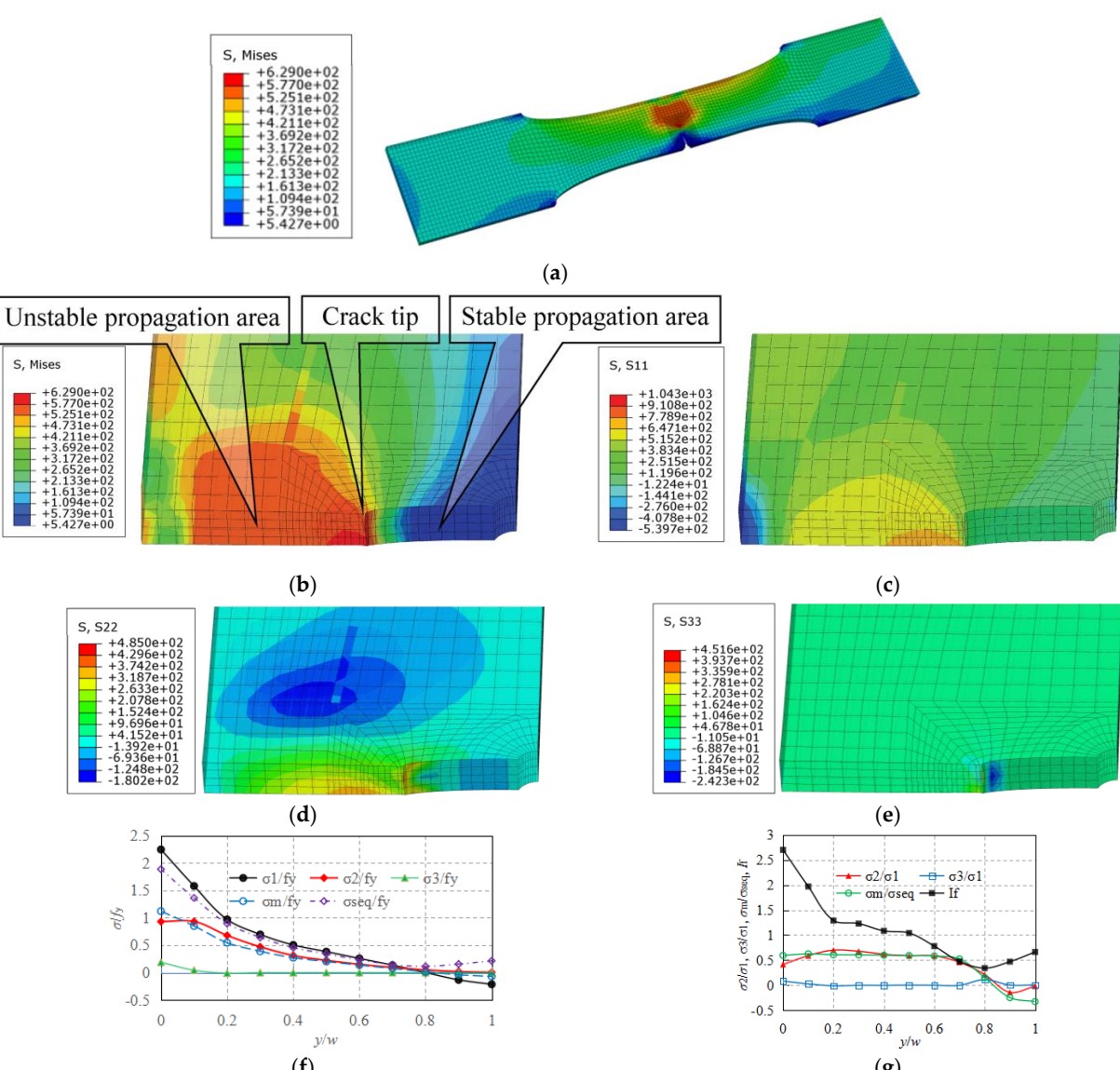

**Figure 6.** Numerically calculated stress field of specimen B1. (**a**) Overall equivalent stress, (**b**) equivalent stress, (**c**) the first stress $\sigma_1$, (**d**) the second stress $\sigma_2$, (**e**) the third stress $\sigma_3$, (**f**) the distribution of relative stress, (**g**) the distribution of stress ratio and $I_f$.

It can be seen in Figure 6a–f that the fatigue crack tip forms a high stress concentration, and the stress field at the crack tip has reached the ellipsoidal fracture model suggested by Equation (3), with the fracture index $I_f = 1.0$, as shown in Figure 6g. A large constraint stress is formed along the thickness of the notched plate at the instability propagation section of the fracture crack. The ratio of the second principle stress to the first principle stress, $\sigma_2/\sigma_1 \approx 0.5$, while the ratio of the third principle stress to the first principle stress, $\sigma_3/\sigma_1 \approx 0.0$, as shown in Figure 6g. The cracking index, $I_f$, at the unstable propagation

section of the fatigue crack sharply increases from the un-notched edge of the notched plate to the crack tip. The peak value of the fracture index, $I_f$, is located at the crack tip. This indicates that the crack tip first cracks and the fatigue crack extends forward along the crack tip. In general, the cracking risk on the unstable propagation area of the fatigue crack is high. Fifty percent of the unstable propagation area of the fatigue crack near the crack tip has cracked. Considering the stress release effect of the crack tip cracking, the unstable propagation area of the fatigue crack will break in the tension.

## 6. Fatigue Life Calculation of Q460C Steel Notched Plates

Table 4 documents the calibrated parameters of the fatigue life calculation model of the Q460C steel notched plates, the unstable propagation area, $A_n$, initiation and stable propagation area, $A_f = A - A_n$, and initiation and stable propagation length, $a_f$, of the fatigue crack in notched plates.

The fatigue life calculation model suggested by Equation (9) is rewritten in double logarithmic coordinates as follows,

$$\log N_f = (\log a_f - \log \xi)/\eta \tag{21}$$

The results of fitting the tested fatigue life, $N_{f,t}$, and numerically calculated stable propagation length, $a_f$, of the fatigue crack in Table 4, according to Equation (21), are shown in Figure 7.

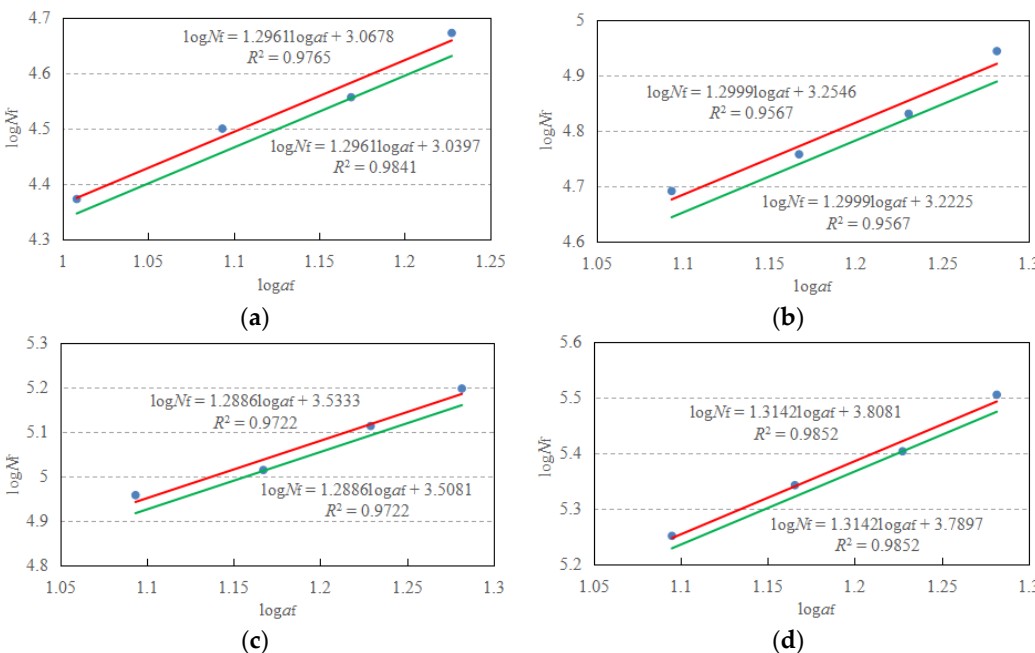

**Figure 7.** Fitting function of the fatigue life of the notched plate and the initiation and stable propagation length of the fatigue crack. (**a**) Specimens B1–B4, (**b**) specimens B5–B8, (**c**) specimens B9–B12, and (**d**) specimens B13–B16.

The fatigue life calculation formula of specimens B1–B4, B5–B8, B9–B12, and B13–B16 are, respectively,

$$\log N_f = 1.2961 \log a_f + 3.0678 \tag{22}$$

$$\log N_f = 1.2999 \log a_f + 3.2546 \tag{23}$$

$$\log N_f = 1.2886 \log a_f + 3.5333 \tag{24}$$

$$\log N_f = 1.3142 \log a_f + 3.8081 \tag{25}$$

From Equations (22)–(25), the calibrated parameters $\xi$ and $\eta$, for specimens B1–B16, together with the standard deviation, $s$, are listed in Table 4.

By introducing 1.645 times the standard deviation, $s$, into Equations (22)–(25), the fatigue life calculation formula of specimens B1–B4, B5–B8, B9–B12, and B13–B16,with 95% assurance rate, are, respectively,

$$\log N_f = 1.2961 \log a_f + 3.0397 \tag{26}$$

$$\log N_f = 1.2999 \log a_f + 3.2225 \tag{27}$$

$$\log N_f = 1.2886 \log a_f + 3.5081 \tag{28}$$

$$\log N_f = 1.3142 \log a_f + 3.7897 \tag{29}$$

From Equations (26)–(29), the calibrated parameter $\xi_{0.95}$, for specimens B1–B16, is listed in Table 4.

**Table 4.** Calibrated parameters of fatigue life calculation model of the notched plates.

| Specimen No. | $A_n$ (mm$^2$) | $A_f$ (mm$^2$) | $a_{f,c}$ (mm) | $\eta$ | $\xi$ ($10^{-3}$) | $s$ | $\xi_{0.95}$ ($10^{-3}$) | $N_{f,c}$ (cycles) | $e_{c\text{-}t}$ (%) |
|---|---|---|---|---|---|---|---|---|---|
| B1 | 73.83 | 41.79 | 10.2 | 0.772 | 4.30 | 0.017 | 4.52 | 23,056 | −6.7 |
| B2 | 62.80 | 49.60 | 12.4 | 0.772 | 4.30 | 0.017 | 4.52 | 29,746 | −5.9 |
| B3 | 55.57 | 60.46 | 14.7 | 0.772 | 4.30 | 0.017 | 4.52 | 37,265 | +3.5 |
| B4 | 45.98 | 69.23 | 16.9 | 0.772 | 4.30 | 0.017 | 4.52 | 44,438 | −1.0 |
| B5 | 64.37 | 50.84 | 12.4 | 0.769 | 3.14 | 0.019 | 3.32 | 47,163 | −3.9 |
| B6 | 54.02 | 58.78 | 14.7 | 0.769 | 3.14 | 0.019 | 3.32 | 58,813 | +2.8 |
| B7 | 46.31 | 69.72 | 17.0 | 0.769 | 3.14 | 0.019 | 3.32 | 71,110 | −1.9 |
| B8 | 36.78 | 78.43 | 19.1 | 0.769 | 3.14 | 0.019 | 3.32 | 82,858 | −5.6 |
| B9 | 63.02 | 49.78 | 12.4 | 0.776 | 1.81 | 0.015 | 1.89 | 81,349 | −10.3 |
| B10 | 55.17 | 60.04 | 14.6 | 0.776 | 1.81 | 0.015 | 1.89 | 101,443 | −1.8 |
| B11 | 45.98 | 69.23 | 16.9 | 0.776 | 1.81 | 0.015 | 1.89 | 122,091 | −5.9 |
| B12 | 36.78 | 78.43 | 19.1 | 0.776 | 1.81 | 0.015 | 1.89 | 142,919 | −9.2 |
| B13 | 64.37 | 50.84 | 12.4 | 0.761 | 1.27 | 0.011 | 1.31 | 159,265 | −10.7 |
| B14 | 54.02 | 58.78 | 14.7 | 0.761 | 1.27 | 0.011 | 1.31 | 196,779 | −10.5 |
| B15 | 46.14 | 69.48 | 16.9 | 0.761 | 1.27 | 0.011 | 1.31 | 236,831 | −6.4 |
| B16 | 35.89 | 76.51 | 19.1 | 0.761 | 1.27 | 0.011 | 1.31 | 278,516 | −12.9 |

As can be seen in Table 4, the parameter $\eta = 0.761$–$0.776$, which is taken conservatively in this paper as, $\eta = 0.77$. The parameter $\xi_{0.95}$ increases with the decrease in relative stress amplitude, $\Delta\sigma/f_y$. The function between the parameter $\xi_{0.95}$ and the relative stress amplitude, $\Delta\sigma/f_y$, shall meet the following conditions:

(1) When $\Delta\sigma/f_y = 0$, the fatigue life $N_f = \infty$. From Equation (9), if $N_f = \infty$, then, $\xi_{0.95} = 0$.

(2) The fatigue life of the Q460 steel notched plate decreases with the increase in relative stress amplitude, $\Delta\sigma/f_y$. From Equation (9), the parameter $\xi_{0.95}$ should be an increasing function of the relative stress amplitude, $\Delta\sigma/f_y$.

The fitting function of the parameters $\xi_{0.95}$ and the relative stress amplitude, $\Delta\sigma/f_y$, that meets the above conditions and has the minimum variance is (Figure 8):

$$\xi_{0.95} = 0.0102 \times (\Delta\sigma/f_y)^{2.3} \tag{30}$$

Substituting in $\eta = 0.77$ and Equation (30) into Equation (9), the fatigue life calculation formula of the Q460 steel notched plates is,

$$N_f = \left(\frac{a_f}{0.0102 \times (\Delta\sigma/f_y)^{2.3}}\right)^{\frac{1}{0.77}} \tag{31}$$

The effect of the relative stress amplitude, $\Delta\sigma/f_y$, and the relative nominal maximum stress, $\sigma_{max}/f_y$, on the fatigue life of the notched plates is taken into account in Equation (31). The fatigue lives calculated by Equation (31), and the relative errors with the test fatigue lives, $e_{c-t} = (N_{f,c} - N_{f,t})/N_{f,t}$, are listed in Table 4. The calculation error of Equation (31) is from $-12.9\%$ to $+5.0\%$, and the calculation accuracy is higher than that of Equation (1). Due to the low sample size of the fatigue test and the large scatter in the tested fatigue life, the results for the fatigue life of specimens B3 and B6 are slightly unsafe, with a margin of error of $+3.5\%$ and $+2.8\%$, respectively. This problem can be addressed by expanding the sample size of fatigue tests.

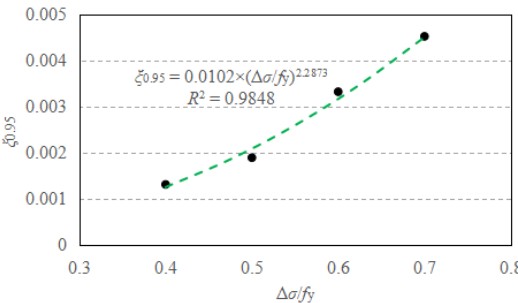

**Figure 8.** Fitting function of the parameter $\xi_{0.95}$ and the relative stress amplitude, $\Delta\sigma/f_y$.

## 7. Conclusions

The propagation of fatigue cracks in tQ460C steel notched plates has been theoretically calculated and numerically simulated, in which the ellipsoidal fracture model was employed as the instability propagation criterion for fatigue cracks. The fatigue life of the Q460C steel notched plates was evaluated using the unified crack growth approach. The conclusions are as follows:

1.　A crack initiates at the edge of the notch, propagates along the width of the plate at the notched section to the un-notched edge of the plate, and penetrates through the thickness of the plate. Fracture occurs at the notched section when the fatigue crack penetrates the width of the notched section.

2.　The relative stress amplitude, $\Delta\sigma/f_y$, rather than the relative converted stress amplitude, $\Delta\sigma_c/f_y$, is the stress parameter that affects the fatigue life of the notched plates.

3.　The fatigue life of the Q460C steel notched plates increases with decreases in the relative stress amplitude, $\Delta\sigma/f_y$, and the relative nominal maximum stress, $\sigma_{max}/f_y$. The effect of the relative stress amplitude, $\Delta\sigma/f_y$, on the fatigue life of the Q460C steel notched plate is larger than that of the relative nominal maximum stress, $\sigma_{max}/f_y$.

4.　The fatigue life predictions determined using the fatigue life formula specified in China's current code are conservative for specimens with a high stress amplitude (i.e., $\Delta\sigma \geq 0.5f_y$), while they are unsafe for those with a low stress amplitude (i.e., $\Delta\sigma = 0.4f_y$), with the calculation errors ranging from $-17.1\%$ to $+84.9\%$.

5.　The fatigue life predictions determined using the modified Gerber ruler are unsafe for the specimens with a high nominal maximum stress (i.e., $\sigma_{max} \geq 0.7f_y$), while they are conservative for those with a low maximum stress (i.e., $\sigma_{max} < 0.7f_y$), with the calculation error being from $-66.8\%$ to $+276.3\%$.

6.　The effects of the relative stress amplitude, $\Delta\sigma/f_y$, and the relative nominal maximum stress, $\sigma_{max}/f_y$, on the fatigue life of the Q460C steel notched plates are considered by means of the unified crack growth approach, which provides prediction errors ranging from $-12.9\%$ to $+3.5\%$. The fatigue life predictions by the unified crack

growth approach are shown to be more accurate, yet generally on the safe side, compared to those predicted according to the method set out in China's current code.

**Author Contributions:** Methodology, W.Z.; formal analysis, F.L.; investigation, W.Z.; data curation, F.L.; writing—original draft preparation, F.L. and W.W.; writing—review and editing, F.L. and W.W.; funding acquisition, W.W. All authors have read and agreed to the published version of the manuscript.

**Funding:** This research was funded by [the Project of Science and Technology for Public Welfare of Ningbo City] grant number [2022S179].

**Data Availability Statement:** The study did not report any data.

**Acknowledgments:** The financial support provided by the Project of Science and Technology for Public Welfare of Ningbo City (Grant no. 2022S179) is gratefully acknowledged. The authors would like to express their sincere thanks for their support.

**Conflicts of Interest:** The authors declare no conflict of interest.

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
