# Peer review of "Fatigue Test and Unified Fatigue Life Calculation of Q460C Steel Notched Plates"

_buildings, doi:10.3390/buildings13030697_

Round 1
Reviewer 1 Report
Manuscript ‘Fatigue test and unified fatigue life calculation of Q460C steel notched plates’
by Fengjun Lv, Wanzhen Wang, and Wei Zhao,
submitted to: Buildings.
This is an intersting paper on the fatigue behaviour of Q460C steel notched plates subjected to high cycle load controlled fatigue. The authors develop a model for this type of test, and validate the model by comparison with actual fatigue test data. The accuracy of the prediction is shown to be higher than that of existing standards.
The research is well designed, and the figures, tables and references are in general adequate. The quality of technical English usage could be improved, and some suggestions in that sense will be given below.
The work follows and enlarges the scope of an earlier work of one of the authors ( ref. [21] Wang, W. Z. (2019) Fatigue failure model of Q345 steel round bars. J. Harb. Inst. Technol., 2019, 51, 58–64). Ref. [21] is not written in English (it is written in Chinese), limiting its potential diffusion and therefore it is usefull and relevant that many considerations concerning modelling are now presented in English.
As technical concerns, the choice of modelling the crack as a cylindrical one ( .... cylindrical fatigue crack with semi-elliptical cross section ..., lines 402-403) instead of the simpler flat crack shape should be explained by the authors.
I offer the following remarks for the consideration of the authors in a possible revision of their manuscript:
Line 70 ‘.... because fatigue fracture exhibits no evident macroscopic plastic deformation. ..’ – this is true for high cycle fatigue, but may be not true for low cycle fatigue. This sentence should be rewritten more carefully.
Line 75 – please write .... SHS (square hollow section) ....
Line 108 – it is ... suitable in general ... ; it is not ... suitable in generally ... .
Line 109 – choice of 0.01 mm for initial crack length deserves some remarks by the present authors.
Line 143 and elsewhere in the manuscript (lines 152, 166, 177, 299, etc. ): ...Where ... is not indented since the authors are not starting a new paragraph.
Line 153 - where it is .... number of cyclic loading. ... should be ... number of load cycles .... . The same applies to line 173, 211.
Line 166 - instead of ... Mises .... please write .... von Mises ... (also in lines 56, 312, 317, 347)
Line 243 - Figure 1 – please improve the quality of the figure – numbers and letters seem distorted ?
Line 255 – it is .... 500 kN ..., it is not .... 500-kN ..... .
Line 272 – do the authors have a photo better focused on the crack surface ?
Line 300 - A = w0 x t.w0 – please check this relation - Is there some typo ?
Lines 402-403 – please justify the geometry of the crack – why is it not just a planar crack?
Limne 495 – why figure with blue background ? uniformity of presentation is preferable.
Author Response
Reviewers' comments:
Reviewer #1:
This is an interesting paper on the fatigue behaviour of Q460C steel notched plates subjected to high cycle load controlled fatigue. The authors develop a model for this type of test, and validate the model by comparison with actual fatigue test data. The accuracy of the prediction is shown to be higher than that of existing standards.
The research is well designed, and the figures, tables and references are in general adequate. The quality of technical English usage could be improved, and some suggestions in that sense will be given below.
The work follows and enlarges the scope of an earlier work of one of the authors ( ref. [21] Wang, W. Z. (2019) Fatigue failure model of Q345 steel round bars. J. Harb. Inst. Technol., 2019, 51, 58–64). Ref. [21] is not written in English (it is written in Chinese), limiting its potential diffusion and therefore it is usefull and relevant that many considerations concerning modelling are now presented in English.
As technical concerns, the choice of modelling the crack as a cylindrical one ( .... cylindrical fatigue crack with semi-elliptical cross section ..., lines 402-403) instead of the simpler flat crack shape should be explained by the authors.
I offer the following remarks for the consideration of the authors in a possible revision of their manuscript:
- 1. As technical concerns, the choice of modelling the crack as a cylindrical one ( ....cylindrical fatigue crack with semi-elliptical cross section..., lines 402-403) instead of the simpler flat crack shape should be explained by the authors.
Response: The reason for modelling the crack as a cylindrical one with semi-elliptical cross section, instead of the simpler flat crack shape, is that the cylindrical fatigue crack with semi-elliptical cross section is easy to simulate the fatigue crack penetrating the thickness of the notch plate, while the simpler flat crack is not easy to simulate the fatigue crack penetrating the thickness of the notch plate, in the ANSYS software.
We have added a detailed explanation based on the reviewer’s suggestion in the revised version as follows: “In order to easily simulate cracks penetrating the thickness of the notched plate in the ANSYS software, cylindrical cracks with semi-elliptic cross sections, rather than the simpler flat crack shape, were introduced at the edges of the notches.”
- Line 70 ‘....because fatigue fracture exhibits no evident macroscopic plastic deformation. ..’ –this is true for high cycle fatigue, but may be not true for low cycle fatigue. This sentence should be rewritten more carefully.
Response: The sentence has been rephrased based on the reviewer’s suggestion as follows: “Since high-cycle fatigue fractures do not exhibit significant macroscopic plastic deformation, sudden failures commonly result in catastrophic accidents and substantial economic losses.”
- Line 75 – please write .... SHS (square hollow section) ....
Response: We have modified this sentence based on the reviewer’s suggestion.
- Line 108 – it is ... suitable in general ... ; it is not ... suitable in generally ....
Response: We have modified this sentence based on the reviewer’s suggestion as follows: “It is not found that the design of the S-N curve in Eurocode3 proved to be suitable in generally, but a sufficient safe stock is not available for this batch of specimens.”
- Line 109 – choice of 0.01 mm for initial crack length deserves some remarks by the present authors.
Response: The initial fatigue crack length is 0.01 mm, depending on the accuracy of the crack observation instrument.
We have revised this sentence as follow: “It should be noted that the initial fatigue crack length is 0.01 mm, depending on the accuracy of the crack observation instrument.”
- Line 143 and elsewhere in the manuscript (lines 152, 166, 177, 299, etc. ): ...Where ... is not indented since the authors are not starting a new paragraph.
Response: We have revised the sentences in line 143, 152, 166, 177, 299, etc., so that they are not indented.
- Line 153 - where it is ....number of cyclic loading.... should be ... number of load cycles .... . The same applies to line 173, 211.
Response: We have corrected lines 153, 173, 211 in for “number of cyclic loading’ to ‘number of load cycles”.
- Line 166 - instead of ... Mises .... please write .... von Mises ... (also in lines 56, 312, 317, 347)
Response: We have corrected lines 166, 563, 312, 317, 347 in for “Mises” to “ von Mises”.
- Line 243 - Figure 1 – please improve the quality of the figure – numbers and letters seem distorted ?
Response: We have corrected Figure 1 to show that the numbers and letters are properly displayed.
- Line 255 – it is .... 500 kN ..., it is not .... 500-kN ..... .
Response: We have corrected lines 255 in for “500-kN” to “500 kN”.
- Line 272 – do the authors have a photo better focused on the crack surface ?
Response: We have added a photo focused on the crack surface in row 272.
- Line 300 - A = w0 x t.w0 – please check this relation - Is there some typo ?
Response: There is no typo in the relation “A = w0 × t”.
- Lines 402-403 – please justify the geometry of the crack – why is it not just a planar crack?
Response: The reason for modelling the crack as a cylindrical one with semi-elliptical cross section, instead of the simpler flat crack shape, is that the cylindrical fatigue crack with semi-elliptical cross section is easy to simulate the fatigue crack penetrating the thickness of the notch plate, while the simpler flat crack is not easy to simulate the fatigue crack penetrating the thickness of the notch plate, in the ANSYS software.
We have added a detailed explanation based on the reviewer’s suggestion in the revised version as follows: “ In order to easily simulate crack penetrating the thickness of the notched plate in the ANSYS software, a cylindrical crack with semi-elliptic cross section, rather than the simpler flat crack shape, is introduced at the edge of the notch.”
- Line 495 – why figure with blue background ? uniformity of presentation is preferable.
Response: We have corrected Figure 4 without the blue background for uniformity of presentation.

Reviewer 2 Report
Dear Authors,
1. Rearrange keywords alphabetically.
2. Figure 1 is not according to the template.
3. Perhaps it would be better to explain in more detail Figure 5, all the images that go into its composition, and if one of them is taken from some other source please cite the source.
4. Some formulas do not respect the templete.
5. Reference for figure 1 needs to be given, and the figure is not included in the text.
6. On same pages of the paper the last line is a subsection/subchapter title, it is not possible that the page ends with a subtitle....
7. The present study's limitations must be added.
8. Mention further research in the conclusion section.
9. Conclusions would be good to be rephrase and extended.
10. Perhaps the list of references should be expanded and checked that all references are included in the text.
As a summary of the review the paper should be reworded a little bit.
After some reworking of the paper it can be resubmitted.
Regards,

Author Response
Reviewers' comments:
Reviewer #2:
- Rearrange keywords alphabetically.
Response: Keywords have been rearranged alphabetically.
- Figure 1 is not according to the template.
Response: We have revised Figure 1 based on the reviewer's suggestion.
- Perhaps it would be better to explain in more detail Figure 5, all the images that go into its composition, and if one of them is taken from some other source please cite the source.
Response: We have added a detailed explanation of Fig. 5 so that all images enter its composition. None of the images in Fig. 5 were taken from some other source.
- Some formulas do not respect the template.
Response to Reviewers We have modified all formulas to respect the template.
- Reference for figure 1 needs to be given and the figure is not included in the text.
Response: Reference [23] for Fig. 1 has been given and Fig. 1 has been included in the text.
- On same pages of the paper the last line is a subsection/subchapter title, it is not possible that the page ends with a subtitle....
Response: We have revised all the pages of the paper so that none of them ends with a subtitle.
- The present study's limitations must be added.
Response: We have included the limitations of the present study in the revised version, as shown in the concluding section as follows:
“It should be noted that the unified fatigue life calculation method presented in this paper can be applied to the fatigue life predictions for metallic materials and structures with small residual stresses. The unified fatigue life calculation method cannot be directly applied to the fatigue life predictions for metal structures with large residual stresses, such as welded joints. The effect of residual stress is not taken into account in the calculation of the stress field at the crack tip and the instability propagation section of the fatigue crack. It is clear that the residual stress has a clear effect on the stress field at the crack tip and on the instability propagation section of the fatigue crack. Further experimental and theoretical studies are needed to calculate fatigue life for metal structures with large residual stresses.”
- Mention further research in the conclusion section.
Response: Further research has been added in the concluding section based on the reviewer’s suggestion.
- Conclusions would be good to be rephrased and extended.
Response: The conclusions have been rephrased and extended based on the reviewer's suggestion.
- Perhaps the list of references should be expanded and checked that all references are included in the text.
Response: References have been expanded and all references are included in the main text, as shown in the references section.

Reviewer 3 Report
The proposal is interesting. The fatigue results from notched specimens with non-zero mean stress cycle are presented.
Introduction part is detailed but based mainly on references from Chinese authors. Theoretical background is based mainly on previous Wang’s works. As some of them are in Chinese the equation derivation can be more detailed.
According to the fatigue test and results there are some issues and non-precise statements:
-The calculated fatigue life in Tabl.3 is based on eq.1 which don’t take into account the mean stress of the loading cycle. Constants Cz and slope b are probably valid for symmetrical (fully reversed) cycle R=-1. This can be the explanation for the registered high errors, especially errors with “+” sign which are on an unsafe side.
Here the Gerber or Goodman rules for taking into account the mean stress can be used.
-Table 4. Derived constraints are declared with 95% assurance rate, but there are two “+” errors from 16 specimens which gives probability 12.5% (2/16 =0.125) more the 2 times higher than 5%. Of course, due to low sample size 5% value can be in the confidence interval limit, but it is better to be mentioned.
There are also some technical mistakes.
- In Fig.1 two types of specimens are shown but only one of them is used.
- Row 284 “specimen C1” is written. May be, should be B1.
- There are some unclear sentences. For example row 124: “In this paper, an experimental study on the fatigue performance of Q460C steel notched plates were investigated”
Recommendation
4 out of 24 references are self-citations , which is a little bit high percent. As a general more references can be included.
English language can be improved.
Author Response
Reviewer #3:
The proposal is interesting. The fatigue results from notched specimens with non-zero mean stress cycle are presented.
Introduction part is detailed but based mainly on references from Chinese authors. Theoretical background is based mainly on previous Wang’s works. As some of them are in Chinese the equation derivation can be more detailed.
According to the fatigue test and results there are some issues and non-precise statements:
- The calculated fatigue life in Table 3 is based on Eq. 1 which doesn’t take into account the mean stress of the loading cycle. Constants Cz and slope b are probably valid for symmetrical (fully reversed) cycle R = -1. This can be the explanation for the registered high errors, especially errors with “+” sign which are on an unsafe side. Here the Gerber or Goodman rules for taking into account the mean stress can be used.
Response: In the revised version, the error analysis of the fatigue life calculated by Eq. (1) in China’s code “Standard for Design of Steel Structures, GB50017-2017” in Table 3 is given as follows:
“The effect of the mean stress of the loading cycle is not taken into account in Eq. (1), resulting in the high errors registered in Table 3, especially those with the “+” sign, which are on the unsafe side. Eq. (1) is valid for symmetrical (fully reversed) cycle R = σmin/σmax = -1. The Gerber or Goodman rules for taking into account the mean stress can be used for unsymmetrical (fully reversed) cycle R ≠ -1.”
- Table 4. Derived constraints are declared with 95% assurance rate, but there are two “+” errors from 16 specimens which gives probability 12.5% (2/16 =0.125) more the 2 times higher than 5%. Of course, due to low sample size 5% value can be in the confidence interval limit, but it is better to be mentioned.
Response: In the revised version we have explained that the calculation of the fatigue life of specimens B3 and B6 in Table 4 is slightly unsafe, with an error of +3.5% and +2.8% respectively, as follows:
“Due to the low sample size of the fatigue test and the large scatter in the test fatigue life, the results for the fatigue life of specimens B3 and B6 are slightly unsafe, with a margin of error of +3.5% and +2.8%, respectively. This problem can be addressed by expanding the sample size of fatigue tests.”
- In Fig.1 two types of specimens are shown but only one of them is used.
Response: We have removed the second type specimen from Fig. 1.
- Row 284 “specimen C1” is written. May be, should be B1.
Response: Row 284 “specimen C1” has been corrected to “specimen B1”.
- There are some unclear sentences. For example row 124: “In this paper, an experimental study on the fatigue performance of Q460C steel notched plates were investigated”
Response: We have revised the sentence as follows: “ In this paper, fatigue tests were carried out on Q460C steel notched plates,…”.
- 4 out of 24 references are self-citations, which is a little bit high percent. As a general more references can be included.
Response: We have removed one reference from the author and added 2 others.

Round 2
Reviewer 2 Report
Dear authors,
The paper needs a few more work.
For example your answer: ”Response: Reference [23] for Fig. 1 has been given” it is not correct because figure 1 does not have reference 23 in the explanation.
At the same time figure 1 looks pretty poor, I don't think it's at a good resolution.
Regards,
Author Response
The paper needs a few more work.
For example your answer: “Response: Reference [23] for Fig. 1 has been given” it is not correct because figure 1 does not have reference 23 in the explanation.
At the same time figure 1 looks pretty poor, I don't think it's at a good resolution.
Response: Reference [20] has been given for Fig. 1 as follows: “Figure 1 shows the geometry of the specimens consist of 20 notched plates of Q460C steel specified in China’s code GB/T 3075-2008 [20].”
We have revised Fig. 1.

Reviewer 3 Report
Dear Authors,
As it was mentioned in my first review, the eq 1 is proper only for the loading cycle R=-1. For your experiment it is not enough only to mention this fakt in the text, but to recalculate the Nf,c in the Table 3. Otherwise the comparison between your equation and eq 1 is not proper.
For example, I recalculate amplitude stress with Gerber formula
dS (Smax) = dS (0) *[ 1- (Smax/fu)^2]
For specimens A1 to A4 the resulting corrected relative amplitudes are:
dS (Smax) = 0.26 ; 0.34; 0.41; 0.48;
Nf,c (Eq. 1) =
973961 |
455925 |
260272 |
163817 |
Al your test results are well bellow the calculated.
Here the stress concentation factor is not considered. You can extract it from FEM static analys and multiply the corrected stress amplitudes.
Of cource you can try the Goodman approximation dS (Smax) = dS (0) *[ 1- (Smax/fu)] and look which is better.
After this recalculations, if your model is still better, conclusion (4) will be supported by the results.
According to Table 4 you can use t-distribution with proper degrees of freedom insted of Normal to calculate Eq 24 -26.
" 1.645 times of standard deviation, s, int"
Author Response
See the file "Reponse to Reviewer 3".

Round 3
Reviewer 3 Report
The new calculations are added to Table 3 and explained.
The conclusion part should be modified a little:
- the results connected with the newly added calculations in Tabl.3 should be mentioned .
The article gives valuable results from non-symmetrical loading fatigue tests and should be accepted with minor corrections.
Author Response
Comments and Suggestions for Authors
- The new calculations are added to Table 3 and explained.
Response: The new calculations based on the Gerber ruler, including the parameters ΔσG, /fy, Nf,G (cycles), and eG-t (%), have been added to Table 3, and the explanations are as follows (see the contents marked in red in the revised version):
“For specimens with large nominal maximum stresses, the stress amplitudes calculated from the Gerber ruler are reduced too much, resulting in large fatigue lives.”
- The conclusion part should be modified a little: The results connected with the newly added calculations in Tabl.3 should be mentioned.
Response: The results connected with the newly added calculations in Tab. 3 have been mentioned in the conclusions as follows (see the conclusion 5 marked in red in the revised version):
“5. The fatigue life predictions determined using the modified Gerber ruler are unsafe for the specimens with high nominal maximum stress (i.e., σmax ≥ 0.7fy) while conservative for those with low maximum stress (i.e., σmax < 0.7fy), with the calculation error is -66.8% ~ +276.3%.”
- The article gives valuable results from non-symmetrical loading fatigue tests and should be accepted with minor corrections.
Response: Many thanks to the kind reviewer.
